# DENOISING IMPROVES LATENT SPACE GEOMETRY IN TEXT AUTOENCODERS

## ABSTRACT

Neural language models have recently shown impressive gains in unconditional text generation, but controllable generation and manipulation of text remain challenging. In particular, controlling text via latent space operations in autoencoders has been difficult, in part due to chaotic latent space geometry. We propose to employ adversarial autoencoders together with denoising (referred as DAAE) to drive the latent space to organize itself. Theoretically, we prove that input sentence perturbations in the denoising approach encourage similar sentences to map to similar latent representations. Empirically, we illustrate the trade-off between text-generation and autoencoder-reconstruction capabilities, and our model significantly improves over other autoencoder variants. Even from completely unsupervised training without style information, DAAE can perform various style transfers, including tense and sentiment, through simple latent vector arithmetic.[1]

## 1 INTRODUCTION

Autoencoder based generative models have recently become popular tools for advancing controllable text generation such as style or sentiment transfer (Bowman et al., 2016; Hu et al., 2017; Shen et al., 2017; Zhao et al., 2018). By mapping sentences to vectors in the latent space, these models offer in principle an attractive, continuous approach to manipulating text by means of simple latent vector arithmetic. However, the success of such manipulations rests heavily on the latent space geometry and how well it agrees with underlying sentence semantics. Indeed, we demonstrate that without additional guidance, fortuitous geometric agreements are unlikely to arise, shedding light on challenges faced by existing methods.

We use adversarial autoencoders (Makhzani et al., 2015, AAEs) to study the latent space geometry. In contrast to variational autoencoders (Kingma & Welling, 2014, VAEs), AAEs can maintain strong coupling between the encoder and decoder that the decoder does not omit the encoded input sentence (Bowman et al., 2016). The training criterion for AAEs consists of two parts, the ability to reconstruct sentences and the additional constraint that the encoded sentences are overall indistinguishable from prior samples, typically Gaussian. We show that these objectives alone do not suffice to force proper latent space geometry for text control. Specifically, for discrete objects such as sentences where continuity assumptions no longer hold, powerful AAEs can easily learn to map training sentences into latent prior samples arbitrarily (Figure 1, Left), while retaining perfect reconstruction. Latent space manipulations in such cases will yield random, unpredictable results.

To remedy this, we augment AAEs with a simple denoising objective (Vincent et al., 2008; Creswell & Bharath, 2018) that requires perturbed sentence with some words missing to be mapped back to the original version. We refer to our model as DAAE. We prove that the denoising criterion can eliminate disorganized solutions and drive the latent space to organize itself. As a result, similar sentences begin to be mapped to similar latent vectors (Figure 1, Right).

Improvements in latent space geometry carry many positive consequences. Through systematic evaluations of the generation and reconstruction capabilities of various text autoencoders (Cífka et al., 2018), we find that our proposed DAAE provides the best trade-off between producing high-quality text vs. informative sentence representations. We empirically verify that DAAE has the best neighborhood preservation property, consistent with our theory. We further investigate to what extent

---

[1] Our code will be made publicly available after the review process.

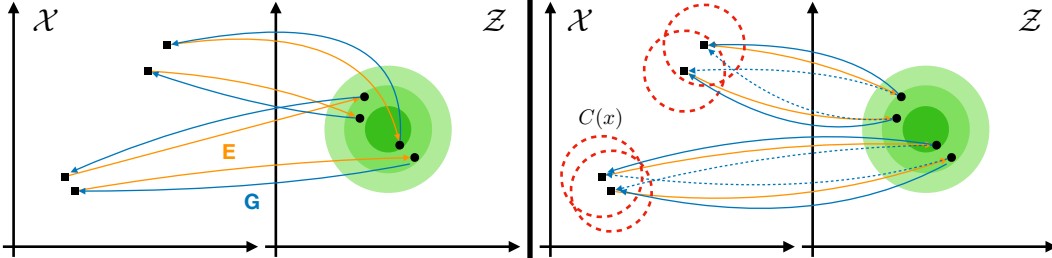

Figure 1: Illustration of the learned latent geometry by AAE before and after introducing $x$ perturbations. With high-capacity encoder/decoder networks, a standard AAE has no preference over $x$-$z$ couplings and thus can learn a random mapping between them (Left). Trained with local perturbations $C(x)$, DAAE learns to map similar $x$ to close $z$ to best achieve the denoising objective (Right).

text can be manipulated by applying simple transformations in the learned latent space. Our model is able to perform sentence-level vector arithmetic (Mikolov et al., 2013) fairly well to change the tense or sentiment of a sentence without any training supervision. It also produces higher quality sentence interpolations than other text autoencoders, suggesting better linguistic continuity in its latent space (Bowman et al., 2016).

## 2 RELATED WORK

Denoising is first introduced into standard autoencoders by Vincent et al. (2008, DAE) to learn robust representations. Without a latent prior, DAE requires sophisticated MCMC sampling to be employed generatively (Bengio et al., 2013). Creswell & Bharath (2018) applied denoising with AAEs to generative image modeling. Here, we demonstrate that input perturbations are particularly useful for discrete text modeling because they encourage preservation of data structure in the latent space.

Apart from the AAE framework that our paper focuses on, another popular latent variable generative model is the variational autoencoder (Kingma & Welling, 2014, VAE). Unfortunately, when the decoder is a powerful autoregressive model (such as a language model), VAE suffers from the *posterior collapse* problem where the latent representations get ignored (Bowman et al., 2016; Chen et al., 2016). If denoising is used in conjunction with VAEs (Im et al., 2017) in text applications, then the noisy inputs will only exacerbate VAE's neglect of the latent variable. Bowman et al. (2016) proposed to weaken VAE's decoder by masking words on the decoder side to alleviate its collapse issue. However, even with a weakened decoder and combined with other techniques including KL-weight annealing and adjusting training dynamics, it is still difficult to inject significant content into the latent code (Yang et al., 2017; Kim et al., 2018; He et al., 2019). Alternatives like the $\beta$-VAE (Higgins et al., 2017) appear necessary.

Previous work on controllable text generation has employed autoencoders trained with attribute label information (Hu et al., 2017; Shen et al., 2017; Zhao et al., 2018; Logeswaran et al., 2018; Subramanian et al., 2018). We show that the proposed DAAE model can perform text manipulations despite being trained in a completely unsupervised manner without attribute labels. This suggests that on the one hand, our model can be adapted to semi-supervised learning when a few labels are available. On the other hand, it can be easily scaled up to train one large model on unlabeled text corpora and then applied for transferring various styles.

## 3 METHOD

Define $\mathcal{X} = \mathcal{V}^m$ to be a space of sequences of discrete symbols from vocabulary $\mathcal{V}$ (with maximum length $m$); also define $\mathcal{Z} = \mathbb{R}^d$ to be a continuous latent space. Our goal is to learn a mapping between the data distribution $p_{\text{data}}(x)$ over $\mathcal{X}$ and a given prior distribution $p(z)$ over latent space $\mathcal{Z}$ (following common practice, a Gaussian prior is used in our experiments, although not required by our methodology). Such a mapping allows us to easily manipulate discrete data through continuous latent representations $z$, and provides a generative model where data samples can be obtained by first drawing $z$ from the prior and then sampling a corresponding sequence via $p(x|z)$.

We adopt the adversarial autoencoder (AAE) framework, which involves a (deterministic) encoder $E : \mathcal{X} \to \mathcal{Z}$, a probabilistic decoder $G : \mathcal{Z} \to \mathcal{X}$, and a discriminator $D : \mathcal{Z} \to [0, 1]$ . Both $E$ and $G$ are recurrent neural networks (RNNs)[2]. $E$ takes input sequence $x$ and outputs the last hidden state as its encoding $z$. $G$ generates a sequence $x$ autoregressively, with each step conditioned on $z$ and previous symbols. The discriminator $D$ is a feed-forward net that outputs the probability of $z$ coming from the prior rather than the encoder. $E$, $G$ and $D$ are trained jointly with a min-max objective:

$$\min_{E,G} \max_{D} \mathcal{L}_{\mathrm{rec}}(\theta_E, \theta_G) - \lambda \mathcal{L}_{\mathrm{adv}}(\theta_E, \theta_D) \tag{1}$$

$$\text{with:} \quad \mathcal{L}_{\mathrm{rec}}(\theta_E, \theta_G) = \mathbb{E}_{p_{\mathrm{data}}(x)}[- \log p_G(x | E(x))] \tag{2}$$

$$\mathcal{L}_{\mathrm{adv}}(\theta_E, \theta_D) = \mathbb{E}_{p(z)}[- \log D(z)] + \mathbb{E}_{p_{\mathrm{data}}(x)}[- \log(1 - D(E(x)))] \tag{3}$$

where reconstruction loss $\mathcal{L}_{\mathrm{rec}}$ and adversarial loss[3] $\mathcal{L}_{\mathrm{adv}}$ are weighted via hyperparameter $\lambda > 0$.

We further introduce perturbations in $\mathcal{X}$ space to learn smoother representations that reflect local structure in the data, ending up with the denoising adversarial autoencoder (DAAE) model. Given a perturbation process $C$ that stochastically maps $x$ to nearby $\tilde{x} \in \mathcal{X}$, let $p(x, \tilde{x}) = p_{\mathrm{data}}(x) p_C(\tilde{x} | x)$ and $p(\tilde{x}) = \sum_x p(x, \tilde{x})$. We change the loss functions to be:

$$\mathcal{L}_{\mathrm{rec}}(\theta_E, \theta_G) = \mathbb{E}_{p(x, \tilde{x})}[- \log p_G(x | E(\tilde{x}))] \tag{4}$$

$$\mathcal{L}_{\mathrm{adv}}(\theta_E, \theta_D) = \mathbb{E}_{p(z)}[- \log D(z)] + \mathbb{E}_{p(\tilde{x})}[- \log(1 - D(E(\tilde{x})))] \tag{5}$$

Here, $\mathcal{L}_{\mathrm{rec}}$ is the loss of reconstructing $x$ from $\tilde{x}$, and $\mathcal{L}_{\mathrm{adv}}$ is the adversarial loss evaluated on *perturbed* $x$. The objective function combines the denoising technique with the AAE (Vincent et al., 2008; Creswell & Bharath, 2018). When $p_C(\tilde{x} | x) = \mathbb{1}[\tilde{x} = x]$ (i.e. there is no perturbation), the above simply becomes the usual AAE objective.

Let $p_E(z | x)$ denote the encoder distribution. With our perturbation process $C$, the posterior distributions of the DAAE are of the following form:

$$q(z | x) = \sum_{\tilde{x}} p_C(\tilde{x} | x) p_E(z | \tilde{x}) \tag{6}$$

This enables the DAAE to utilize stochastic encodings even by merely employing a deterministic encoder network trained without any reparameterization-style tricks. Note that since $q(z | x)$ of the form (6) is a subset of all possible conditional distributions, our model is still minimizing an upper bound of the Wasserstein distance between data and model distributions, as previously shown by Tolstikhin et al. (2017) for AAE (see Appendix A for a full proof).

## 4  LATENT SPACE GEOMETRY

The latent space geometry of text autoencoders is an important yet understudied problem. Only when the latent space is smooth and regular can meaningful text manipulations be enacted via simple modifications of the corresponding latent representations. Here, we discuss in detail the posterior characteristics of the DAAE, and provide a theoretical analysis of how input perturbations help better structure the latent space geometry (all proofs are relegated to the appendix).

Assume our perturbations preserve $x$ with some probability (i.e. $p_C(x | x) > 0$). When the support of $C(x_1)$ and $C(x_2)$ do not overlap for different training examples $x_1 \neq x_2$, the encoder can learn to assign $p_E(z | \tilde{x}) = p_E(z | x)$ for $\tilde{x} \in C(x)$, and we are back to the unconstrained posterior scenario $q(z | x) = p_E(z | x)$ (Eq. 6). If $C(x_1)$ and $C(x_2)$ do intersect, then the latent posterior of $x_1$ and $x_2$ will have overlapping components $p_E(z | \tilde{x})$ for $\tilde{x} \in C(x_1) \cap C(x_2)$. For example, if $p_C(\tilde{x} | x)$ assigns a high probability to $\tilde{x}$ that lies close to $x$ (based on some metric over $\mathcal{X}$), then for similar $x_1$ and $x_2$, the high-probability overlap between their perturbations will inherently force their posteriors closer together in the latent space. This is desirable for learning good representations $z$, while not guaranteed by merely minimizing the statistical divergence between $p_{\mathrm{data}}(x)$ and $p_G(x) = \mathbb{E}_{p(z)}[p_G(x | z)]$.

Now we formally analyze which type of $x$-$z$ mappings will be learned by AAE and DAAE, respectively, to achieve global optimality of their training objectives. Unlike previous analyses of

---

[2]Transformer models (Vaswani et al., 2017) did not outperform LSTMs on our moderately-sized datasets.

[3]We actually train $E$ to maximize $\log D(E(x))$ instead of $- \log(1 - D(E(x)))$, which is more stable in practice (Goodfellow et al., 2014). We also tried WGAN (Arjovsky et al., 2017) but did not notice any gains.

noise in single-layer networks (Poole et al., 2014), here we study high-capacity encoder/decoder networks (Schäfer & Zimmermann, 2006) with a large number of parameters that are used in modern sequence models (Devlin et al., 2018; Radford et al., 2019). Throughout, we assume that:

**Assumption 1.** *E is a universal approximator capable of producing any mapping from $x$'s to $z$'s.*

**Assumption 2.** *G can approximate arbitrary $p(x|z)$ so long as it remains sufficiently Lipschitz continuous in $z$. Namely, there exists $L > 0$ such that all decoder models $G$ obtainable via training satisfy that for all $x \in \mathcal{X}, z_1, z_2 \in \mathcal{Z}$: $|\log p_G(x|z_1) - \log p_G(x|z_2)| \leq L\|z_1 - z_2\|$.*

Following prior analysis of language decoders (Mueller et al., 2017), we assume that $G$ is $L$-Lipschitz in its continuous input $z$ (denote this set of possible decoders by $\mathcal{G}_L$). When $G$ is implemented as a RNN or Transformer language model, $\log p_G(x|z)$ will remain Lipschitz in $z$ if the recurrent or attention weight matrices have bounded norm. This property is naturally encouraged by popular training methods that utilize SGD with early stopping and $L_2$ regularization (Zhang et al., 2017). Note we have not assumed $E$ or $G$ is Lipschitz in $x$, which would be unreasonable since $x$ stands for discrete text, and when a few symbols change, the decoder likelihood for the entire sequence can vary drastically (e.g., $G$ may assign a much higher probability to a grammatical sentence than an ungrammatical one that only differs by one word). Our discussion is directed to the nature of such families of log-likelihood functions with a continuous variable $z$ and a discrete variable $x$.

We further assume an effectively trained discriminator that succeeds in its adversarial task:

**Assumption 3.** *D ensures that the latent encodings $z_1, \cdots, z_n$ of training examples $x_1, \cdots, x_n$ are indistinguishable from prior samples $z \sim p(z)$.*

For simplicity, we directly assume that $z_1, \cdots, z_n$ are actual samples from $p(z)$ which are given a priori. Here, the task of the encoder $E$ is to map given unique training examples to the given latent points, and the goal of the decoder $p_G(\cdot|\cdot)$ is to maximize $-\mathcal{L}_{rec}$ under the encoder mapping. The question now is which one-to-one mapping an optimal encoder/decoder will learn under the AAE and DAAE objective (Eq. 2 and Eq. 4). We start with the following observation:

**Theorem 1.** *For any one-to-one encoder mapping $E$ from $\{x_1, \cdots, x_n\}$ to $\{z_1, \cdots, z_n\}$, the optimal value of objective $\max_{G \in \mathcal{G}_L} \frac{1}{n} \sum_{i=1}^{n} \log p_G(x_i|E(x_i))$ is the same.*

Intuitively, this result stems from the fact that the model receives no information about the structure of $x$, and $x_1, \cdots, x_n$ are simply provided as different symbols. Hence AAE offers no preference over $x$-$z$ couplings, and a random matching in which the $z$ do not reflect any data structure is equally good as any other matching (Figure 1, Left). Latent point assignments start to differentiate, however, once we introduce local input perturbations.

To elucidate how perturbations affect latent space geometry, it helps to first consider a simple setting with only four examples $x_1, x_2, x_3, x_4 \in \mathcal{X}$. Again, we consider given latent points $z_1, z_2, z_3, z_4$ sampled from $p(z)$, and the encoder/decoder are tasked with learning which $x$ to match with which $z$. As depicted in Figure 1, suppose there are two pairs of $x$ closer together and also two pairs of $z$ closer together. Let $\sigma$ denote the sigmoid function, we have the following conclusion:

**Theorem 2.** *Let $d$ be a distance metric over $\mathcal{X}$. Suppose $x_1, x_2, x_3, x_4$ satisfy that with some $\epsilon > 0$: $d(x_1, x_2) < \epsilon$, $d(x_3, x_4) < \epsilon$, and $d(x_i, x_j) > \epsilon$ for all other $(x_i, x_j)$ pairs. In addition, $z_1, z_2, z_3, z_4$ satisfy that with some $0 < \delta < \zeta$: $\|z_1 - z_2\| < \delta$, $\|z_3 - z_4\| < \delta$, and $\|z_i - z_j\| > \zeta$ for all other $(z_i, z_j)$ pairs. Suppose our perturbation process $C$ reflects local $\mathcal{X}$ geometry with: $p_C(x_i|x_j) = 1/2$ if $d(x_i, x_j) < \epsilon$ and $= 0$ otherwise. For $\delta < \frac{1}{L} (2 \log (\sigma(L\zeta)) + \log 2)$ and $\zeta > \frac{1}{L} \log (1/(\sqrt{2} - 1))$, the denoising objective $\max_{G \in \mathcal{G}_L} \frac{1}{n} \sum_{i=1}^{n} \sum_{j=1}^{n} p_C(x_j|x_i) \log p_G(x_i|E(x_j))$ (where $n = 4$) achieves the largest value when encoder $E$ maps close pairs of $x$ to close pairs of $z$.*

This entails that DAAE will always prefer to map similar $x$ to similar $z$. Note that Theorem 1 still applies here, and AAE will not prefer any particular $x$-$z$ pairing over the other possibilities. We next generalize beyond the basic four-points scenario to consider $n$ examples of $x$ that are clustered. Here, we can ask whether this cluster organization will be reflected in the latent space of DAAE.

**Theorem 3.** *Suppose $x_1, \cdots, x_n$ are divided into $n/K$ clusters of equal size $K$, with $S_i$ denoting the cluster index of $x_i$. Let the perturbation process $C$ be uniform within clusters, i.e. $p_C(x_i|x_j) = 1/K$ if $S_i = S_j$ and $= 0$ otherwise. For a one-to-one encoder mapping $E$ from $\{x_1, \cdots, x_n\}$ to $\{z_1, \cdots, z_n\}$, the denoising objective $\max_{G \in \mathcal{G}_L} \frac{1}{n} \sum_{i=1}^{n} \sum_{j=1}^{n} p_C(x_j|x_i) \log p_G(x_i|E(x_j))$ is upper bounded by: $\frac{1}{n^2} \sum_{i,j:S_i \neq S_j} \log \sigma(L\|E(x_i) - E(x_j)\|) - \log K$.*

Theorem 3 provides an upper bound of the DAAE objective that can be achieved by a particular $x$-$z$ mapping. This achievable limit is substantially better when examples in the same cluster are mapped to the latent space in a manner that is well-separated from encodings of other clusters. In other words, by preserving input space cluster structure in the latent space, DAAE can achieve better objective values and thus is incentivized to learn such encoder/decoder mappings. An analogous corollary can be shown for the case when examples $x$ are perturbed to yield additional inputs $\tilde{x}$ not present in the training data. In this case, the model would aim to map each example and its perturbations as a group to a compact group of $z$ points well-separated from other groups in the latent space.

In conclusion, our analysis shows that a well-trained DAAE is guaranteed to learn neighborhood-preserving latent representations, whereas even a perfectly-trained AAE model may learn latent representations whose geometry fails to reflect similarity in the $x$ space. Empirical experiments in Section 5.2 confirm that our theory holds in practice.

## 5 EXPERIMENTS

We evaluate our proposed model and other text autoencoders on two benchmark datasets: *Yelp reviews* and *Yahoo answers* (Shen et al., 2017; Yang et al., 2017). Detailed descriptions of datasets, training settings, human evaluations, and additional results/examples can be found in the appendix.

**Perturbation Process** We randomly delete each word with probability $p$, so that perturbations of sentences with more words in common will have a larger overlap. We also tried replacing each word with a <mask> token or a random word and found that they all brought improvements, but deleting words worked best. We leave it to future work to explore more sophisticated text perturbations.

**Baselines** We compare our proposed DAAE with four alternative text autoencoders: adversarially regularized autoencoder (Zhao et al., 2018, ARAE), $\beta$-VAE (Higgins et al., 2017), AAE (Makhzani et al., 2015), and latent-noising AAE (Rubenstein et al., 2018, LAAE). Similar to our model, the LAAE uses Gaussian perturbations in the latent space to improve AAE's latent geometry (rather than perturbations in the sentence space). However, LAAE requires enforcing an $L_1$ penalty ($\lambda_1 \cdot \| \log \sigma^2(x) \|_1$) on the latent perturbations' log-variance to prevent them from vanishing. In contrast, input perturbations in DAAE enable stochastic latent representations without parametric restrictions like Gaussianity.

### 5.1 GENERATION-RECONSTRUCTION TRADE-OFF

We evaluate various latent variable generative models in terms of both generation quality and reconstruction accuracy. A strong model should not only generate high quality sentences, but also learn useful latent variables that capture significant data content. Recent work on text autoencoders has found an inherent tension between these aims (Bowman et al., 2016; Cífka et al., 2018), yet only when both goals are met can we successfully manipulate sentences by modifying their latent representation (in order to produce valid output sentences that retain the semantics of the input).

We compute the BLEU score (Papineni et al., 2002) between input and reconstructed sentences to measure reconstruction accuracy, and compute Forward/Reverse PPL to measure sentence generation quality (Zhao et al., 2018; Cífka et al., 2018).[4] Forward PPL is the perplexity of a language model trained on real data and evaluated on generated data. It measures the fluency of the generated text, but cannot detect the collapsed case where the model repeatedly generates a few common sentences. Reverse PPL is the perplexity of a language model trained on generated data and evaluated on real data. It takes into account both the fluency and diversity of the generated text. If a model generates only a few common sentences, a language model trained on it will exhibit poor PPL on real data.

We thoroughly investigate the performance of different models and their trade-off between generation and reconstruction. Figure 2 plots reconstruction BLEU (higher is better) vs. Forward/Reverse PPL (lower is better). The lower right corner indicates an ideal situation where good reconstruction accuracy and generation quality are both achieved. For models with tunable hyperparameters, we sweep the full spectrum of their generation-reconstruction trade-off by varying the KL coefficient $\beta$ of $\beta$-VAE, the log-variance $L_1$ penalty $\lambda_1$ of LAAE, and the word drop probability $p$ of DAAE.

---

[4] While some use importance sampling estimates of data likelihood to evaluate VAEs (He et al., 2019), adopting the encoder as a proposal density is not suited for AAE variants, as they are optimized based on Wasserstein distances rather than likelihoods and lack closed-form posteriors.

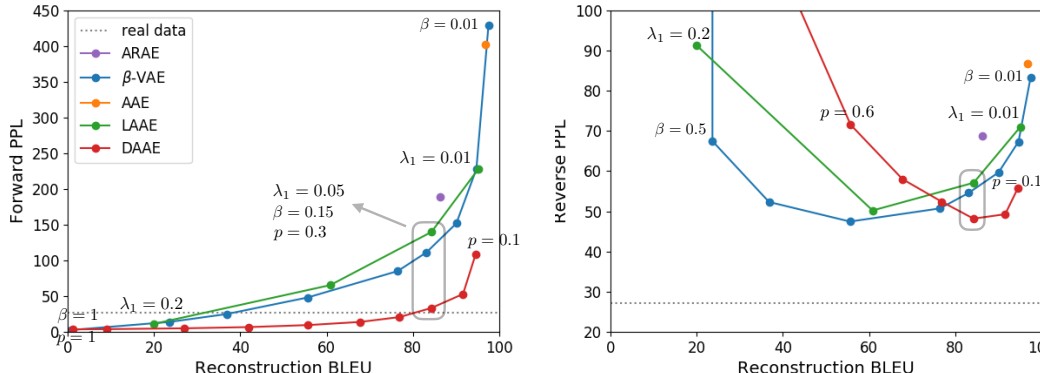

Figure 2: Generation-reconstruction trade-off of various text autoencoders on Yelp. The "real data" line marks the PPL of a language model trained and evaluated on real data. We strive to approach the lower right corner with both high BLEU and low PPL. The grey box identifies hyperparameters we use for respective models in subsequent experiments. Points of severe collapse (Reverse PPL > 200) are removed from the right panel.

In the left panel, we observe that a standard VAE ($\beta = 1$) completely collapses and ignores the latent variable $z$, resulting in reconstruction BLEU close to 0. At the other extreme, AAE can achieve near-perfect reconstruction, but its latent space is highly non-smooth and generated sentences are of poor quality, indicated by its large Forward PPL. Decreasing $\beta$ in VAE or introducing latent noises in AAE provides the model with a similar trade-off curve between reconstruction and generation. We note that ARAE falls on or above their curves, revealing that it does not fare better than these methods (Cífka et al. (2018) also reported similar findings). Our proposed DAAE provides a trade-off curve that is strictly superior to other models. With discrete $x$ and a complex encoder, the Gaussian perturbations added to the latent space by $\beta$-VAE and LAAE are not directly related to how the inputs are encoded. In contrast, input perturbations added by DAAE can constrain the encoder to maintain coherence between neighboring inputs in an end-to-end fashion and help learn smoother latent space.

The right panel in Figure 2 illustrates that Reverse PPL first drops and then rises as we increase the degree of regularization/perturbation. This is because when $z$ encodes little information, generations from prior-sampled $z$ lack enough diversity to cover the real data. Again, DAAE outperforms the other models which tend to have higher Reverse PPL and lower reconstruction BLEU. In subsequent experiments, we set $\beta = 0.15$ for $\beta$-VAE, $\lambda_1 = 0.05$ for LAAE, and $p = 0.3$ for DAAE, to ensure they have strong reconstruction abilities and encode enough information to enable text manipulations.

## 5.2 Neighborhood Preservation

In this section, we empirically investigate whether our previous theory holds in practice. That is, in actual autoencoder models trained on real text datasets, do sentence perturbations induce latent space organization that better preserves neighborhood structure in the data space?

Under our word-drop perturbation process, sentences with more words in common are more likely to be perturbed into one another. This choice of $C$ approximately encodes sentence similarity via the normalized edit distance[5]. Within the test set, we find both the 10 nearest neighbors of each sentence based on the normalized edit distance (denote this set by $\text{NN}_x$), as well as the $k$ nearest neighbors based on Euclidean distance between latent representations (denote this set by $\text{NN}_z$). We

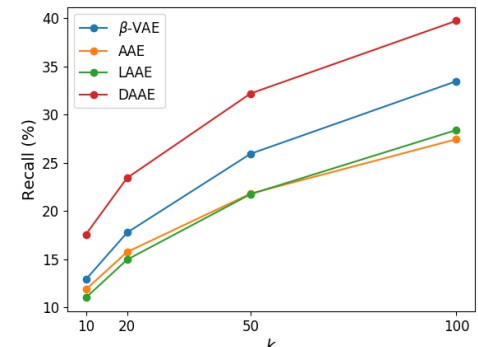

Figure 3: Recall rate of 10 nearest neighbors in the sentence space retrieved by $k$ nearest neighbors in the latent space on Yelp. ARAE is not plotted here as we find its recall significantly below other models ($< 1\%$).

---

[5]Normalized edit distance $\in [0, 1]$ is the Levenshtein distance divided by the max length of two sentences.

|  | AAE | DAAE |
|---|---|---|
| **Source** | **my waitress katie was fantastic , attentive and personable .** | **my waitress katie was fantastic , attentive and personable .** |
|  | my cashier did not smile , barely said hello . | the manager , linda , was very very attentive and personable . |
|  | the service is fantastic , the food is great . | stylist brenda was very friendly , attentive and professional . |
|  | the employees are extremely nice and helpful . | the manager was also super nice and personable . |
|  | our server kaitlyn was also very attentive and pleasant . | my server alicia was so sweet and attentive . |
|  | the crab po boy was also bland and forgettable . | our waitress ms. taylor was amazing and very knowledgeable . |
| **Source** | **i have been known to eat two meals a day here .** | **i have been known to eat two meals a day here .** |
|  | i have eaten here for _num_ years and never had a bad meal ever . | you can seriously eat one meal a day here . |
|  | i love this joint . | i was really pleased with our experience here . |
|  | i have no desire to ever have it again . | ive been coming here for years and always have a good experience . |
|  | you do n't need to have every possible dish on the menu . | i have gone to this place for happy hour for years . |
|  | i love this arena . | we had _num_ ayce dinner buffets for _num_ on a tuesday night . |

Table 1: Examples of 5 nearest neighbors in the latent Euclidean space of AAE and DAAE on the Yelp dataset.

| Model | ACC | BLEU | PPL |
|---|---|---|---|
| ARAE | 17.2 | 55.7 | 59.1 |
| $\beta$-VAE | 49.0 | 43.5 | 44.4 |
| AAE | 9.7 | **82.2** | 37.4 |
| LAAE | 43.6 | 37.5 | 55.8 |
| DAAE | **50.3** | 54.3 | **32.0** |

| $\beta$-VAE is better: 25 | DAAE is better: **48** |
|---|---|
| both good: 26 | both bad: 67     n/a: 34 |

Table 2: Above: automatic evaluations of vector arithmetic for tense inversion. Below: human evaluation statistics of our model vs. the closest baseline $\beta$-VAE.

| Model |  | ACC | BLEU | PPL |
|---|---|---|---|---|
| Shen et al. (2017) |  | 81.7 | 12.4 | 38.4 |
| AAE | $\pm v$ | 7.2 | 86.0 | 33.7 |
|  | $\pm 1.5v$ | 25.1 | 59.6 | 59.5 |
|  | $\pm 2v$ | 57.5 | 27.4 | 139.8 |
| DAAE | $\pm v$ | 36.2 | 40.9 | 40.0 |
|  | $\pm 1.5v$ | 73.6 | 18.2 | 54.1 |
|  | $\pm 2v$ | 91.8 | 7.3 | 61.8 |

Table 3: Automatic evaluations of vector arithmetic for sentiment transfer. Accuracy (ACC) is measured by a sentiment classifier. The model of Shen et al. (2017) is specifically trained for sentiment transfer with labeled data, while our text autoencoders are not.

compute the recall rate $|\text{NN}_x \cap \text{NN}_z| \, / \, |\text{NN}_x|$, which indicates how well local neighborhoods are preserved in the latent space of different models.

Figures 3 shows that DAAE consistently gives the highest recall, about 1.5∼2 times that of AAE, implying that input perturbations have a substantial effect on shaping the latent space geometry. Tables 1 presents the five nearest neighbors found by AAE and DAAE in their latent space for example test set sentences. The AAE sometimes encodes entirely unrelated sentences close together, while the latent space geometry of the DAAE is structured based on key words such as "attentive" and "personable", and tends to group sentences with similar semantics close together.

## 5.3 APPLICATIONS TO CONTROLLABLE TEXT GENERATION

### 5.3.1 STYLE TRANSFER VIA VECTOR ARITHMETIC

Mikolov et al. (2013) previously discovered that word embeddings from unsupervised learning can capture linguistic relationships via simple arithmetic. A canonical example is the embedding arithmetic "King" - "Man" + "Woman" ≈ "Queen". Here, we use the Yelp dataset with tense and sentiment as two example attributes (Hu et al., 2017; Shen et al., 2017) to investigate whether analogous structure emerges in the latent space of our sentence-level models.

**Tense** We use the Stanford Parser to extract the main verb of a sentence and determine the sentence tense based on its part-of-speech tag. We compute a single "tense vector" by averaging the latent code $z$ separately for 100 past tense sentences and 100 present tense sentences in the dev set, and then calculating the difference between the two. Given a sentence from the test set, we attempt to change its tense from past to present or from present to past through simple addition/subtraction of the tense vector. More precisely, a source sentence $x$ is first is encoded to $z = E(x)$, and then the tense-modified sentence is produced via $G(z \pm v)$, where $v \in \mathbb{R}^d$ denotes the fixed tense vector.

| Input | i enjoy hanging out in their hookah lounge . | had they informed me of the charge i would n't have waited . |
|---|---|---|
| ARAE | i enjoy hanging out in their 25th lounge . | amazing egg of the may i actually ! |
| $\beta$-VAE | i made up out in the backyard springs salad . | had they help me of the charge i would n't have waited . |
| AAE | i enjoy hanging out in their brooklyn lounge . | have they informed me of the charge i would n't have waited . |
| LAAE | i enjoy hanging out in the customized and play . | they are girl ( the number so i would n't be forever . |
| DAAE | i enjoyed hanging out in their hookah lounge . | they have informed me of the charge i have n't waited . |

Table 4: Examples of vector arithmetic for tense inversion.

| | AAE | DAAE |
|---|---|---|
| Input | the food is entirely tasteless and slimy . | the food is entirely tasteless and slimy . |
| $+v$ | the food is entirely tasteless and slimy . | the food is tremendous and fresh . |
| $+1.5v$ | the food is entirely tasteless and slimy . | the food is sensational and fresh . |
| $+2v$ | the food is entirely and beef . | the food is gigantic . |
| Input | i really love the authentic food and will come back again . | i really love the authentic food and will come back again . |
| $-v$ | i really love the authentic food and will come back again . | i really love the authentic food and will never come back again . |
| $-1.5v$ | i really but the authentic food and will come back again . | i really do not like the food and will never come back again . |
| $-2v$ | i really but the worst food but will never come back again . | i really did not believe the pretentious service and will never go back . |

Table 5: Examples of vector arithmetic for sentiment transfer.

To quantitatively compare different models, we compute their tense transfer accuracy as measured by the parser, the output BLEU with the input sentence, and output (forward) PPL evaluated by a language model. DAAE achieves the highest accuracy, lowest PPL, and relatively high BLEU (Table 2, Above), indicating that the output sentences produced by our model are more likely to be of high quality and of the proper tense, while remaining similar to the source sentence. A human evaluation on 200 test sentences (100 past and 100 present, details in Appendix G) suggests that DAAE outperforms $\beta$-VAE twice as often as it is outperformed, and our model successfully inverts tense for $(48 + 26)/(200 - 34) = 44.6\%$ of sentences, 13.8% more than $\beta$-VAE (Table 2, Below). Tables 4 and J.2 show the results of adding or subtracting this fixed latent vector offset under different models. DAAE can successfully change "enjoy" to "enjoyed", or change the subjunctive mood to declarative mood and adjust the word order. Other baselines either fail to alter the tense, or undesirably change the semantic meaning of the source sentence (e.g. "enjoy" to "made").

**Sentiment** Following the same procedure used to alter tense, we compute a "sentiment vector" $v$ from 100 negative and positive sentences and use it to change the sentiment of test sentences. Table 3 reports the automatic evaluations, and Tables 5 and J.3 show examples generated by AAE and DAAE. Scaling $\pm v$ to $\pm 1.5v$ and $\pm 2v$, we find that the resulting sentences get more and more positive/negative. However, the PPL for AAE increases rapidly with this scaling factor, indicating that the sentences become unnatural when their encodings have a large offset. DAAE enjoys a much smoother latent space than AAE. Despite the fact that no sentiment labels are provided during training (a more challenging task than previous style transfer models (Shen et al., 2017)), DAAE with $\pm 1.5v$ is able to transfer sentiment fairly well.

### 5.3.2 SENTENCE INTERPOLATION VIA LATENT SPACE TRAVERSAL

We also study sentence interpolation by traversing the latent space of text autoencoders. Given two input sentences, we encode them to $z_1, z_2$ and decode from $tz_1 + (1 - t)z_2$ $(0 \leq t \leq 1)$ to obtain interpolated sentences. Ideally this should produce fluent sentences with gradual semantic change (Bowman et al., 2016). Table 6 shows two examples from the Yelp dataset, where it is clear that DAAE produces more coherent and natural interpolations than AAE. Table J.4 in the appendix shows two difficult examples from the Yahoo dataset, where we interpolate between dissimilar sentences. While it is challenging to generate semantically correct sentences in these cases, the latent space of our model exhibits continuity on topic and syntactic structure.

## 6 CONCLUSION

This paper proposed DAAE for generative text modeling. As revealed in previous work (Devlin et al., 2018; Lample et al., 2018), we find that denoising techniques can greatly improve the learned text representations. We provide a theoretical explanation for this phenomenon by analyzing the latent

| | | |
|---|---|---|
| **Input 1** | **it 's so much better than the other chinese food places in this area .** | **fried dumplings are a must .** |
| **Input 2** | **better than other places .** | **the fried dumplings are a must if you ever visit this place .** |
| AAE | it 's so much better than the other chinese food places in this area . | fried dumplings are a must . |
| | it 's so much better than the other food places in this area . | fried dumplings are a must . |
| | better , much better . | the dumplings are a must if you worst . |
| | better than other places . | the fried dumplings are a must if you ever this place . |
| | better than other places . | the fried dumplings are a must if you ever visit this place . |
| DAAE | it 's so much better than the other chinese food places in this area . | fried dumplings are a must . |
| | it 's much better than the other chinese places in this area . | fried dumplings are a must visit . |
| | better than the other chinese places in this area . | fried dumplings are a must in this place . |
| | better than the other places in charlotte . | the fried dumplings are a must we ever visit this . |
| | better than other places . | the fried dumplings are a must if we ever visit this place . |

Table 6: Interpolations between two input sentences generated by AAE and our model on the Yelp dataset.

space geometry arisen from input perturbations. Our proposed model substantially outperforms other text autoencoders, and demonstrates potential for various text manipulations via vector operations. Future work may investigate superior perturbation strategies and additional properties of latent space geometry to provide finer control over the text generated using autoencoder models.

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

## A    WASSERSTEIN DISTANCE

The AAE objective can be connected to a relaxed form of the Wasserstein distance between model and data distributions (Tolstikhin et al., 2017). Specifically, for cost function $c(\cdot, \cdot) : \mathcal{X} \times \mathcal{X} \to \mathbb{R}$ and deterministic decoder mapping $G : \mathcal{Z} \to \mathcal{X}$, it holds that:

$$\inf_{\Gamma \in \mathcal{P}(x \sim p_{\text{data}}, y \sim p_G)} \mathbb{E}_{(x,y) \sim \Gamma}[c(x,y)] = \inf_{q(z|x):q(z)=p(z)} \mathbb{E}_{p_{\text{data}}(x)} \mathbb{E}_{q(z|x)}[c(x, G(z))] \tag{7}$$

where the minimization over couplings $\Gamma$ with marginals $p_{\text{data}}$ and $p_G$ can be replaced with minimization over conditional distributions $q(z|x)$ whose marginal $q(z) = \mathbb{E}_{p_{\text{data}}(x)}[q(z|x)]$ matches the latent prior distribution $p(z)$. Relaxing this marginal constraint via a divergence penalty $D(q(z)\|p(z))$ estimated by adversarial training, one recovers the AAE objective (Eq. 1). In particular, AAE on discrete $x$ with the cross-entropy loss is minimizing an upper bound of the total variation distance between $p_{\text{data}}$ and $p_G$, with $c$ chosen as the indicator cost function (Zhao et al., 2018).

Our model is optimizing over conditional distributions $q(z|x)$ of the form (6), a subset of all possible conditional distributions. Thus, after introducing input perturbations, our method is still minimizing an upper bound of the Wasserstein distance between $p_{\text{data}}$ and $p_G$ described in (7).

## B    PROOF OF THEOREM 1

**Theorem 1.** *For any one-to-one encoder mapping $E$ from $\{x_1, \cdots, x_n\}$ to $\{z_1, \cdots, z_n\}$, the optimal value of objective $\max_{G \in \mathcal{G}_L} \frac{1}{n} \sum_{i=1}^{n} \log p_G(x_i|E(x_i))$ is the same.*

*Proof.* Consider two encoder matchings $x_i$ to $z_{\alpha(i)}$ and $x_i$ to $z_{\beta(i)}$, where both $\alpha$ and $\beta$ are permutations of the indices $\{1, \ldots, n\}$. Suppose $G_\alpha$ is the optimal decoder model for the first matching (with permutations $\alpha$). This implies

$$p_{G_\alpha} = \arg\max_{G \in \mathcal{G}_L} \sum_{i=1}^{n} \log p_G(x_i|z_{\alpha(i)})$$

Now let $p_{G_\beta}(x_i|z_j) = p_{G_\alpha}(x_{\beta\alpha^{-1}(i)}|z_j), \forall i, j$. Then $G_\beta$ can achieve exactly the same log-likelihood objective value for matching $\beta$ as $G_\alpha$ for matching $\alpha$, while still respecting the Lipschitz constraint. ☐

## C    PROOF OF THEOREM 2

**Theorem 2.** *Let $d$ be a distance metric over $\mathcal{X}$. Suppose $x_1, x_2, x_3, x_4$ satisfy that with some $\epsilon > 0$: $d(x_1, x_2) < \epsilon$, $d(x_3, x_4) < \epsilon$, and $d(x_i, x_j) > \epsilon$ for all other $(x_i, x_j)$ pairs. In addition, $z_1, z_2, z_3, z_4$ satisfy that with some $0 < \delta < \zeta$: $\|z_1 - z_2\| < \delta$, $\|z_3 - z_4\| < \delta$, and $\|z_i - z_j\| > \zeta$ for all other $(z_i, z_j)$ pairs. Suppose our perturbation process $C$ reflects local $\mathcal{X}$ geometry with: $p_C(x_i|x_j) = 1/2$ if $d(x_i, x_j) < \epsilon$ and $= 0$ otherwise. For $\delta < \frac{1}{L}\left(2\log\left(\sigma(L\zeta)\right) + \log 2\right)$ and $\zeta > \frac{1}{L}\log\left(1/(\sqrt{2} - 1)\right)$, the denoising objective $\max_{G \in \mathcal{G}_L} \frac{1}{n} \sum_{i=1}^{n} \sum_{j=1}^{n} p_C(x_j|x_i) \log p_G(x_i|E(x_j))$ (where $n = 4$) achieves the largest value when encoder $E$ maps close pairs of $x$ to close pairs of $z$.*

*Proof.* Let $[n]$ denote $\{1, \ldots, n\}$, and assume without loss of generality that the encoder $E$ maps each $x_i$ to $z_i$. We also define $A = \{1, 2\}, B = \{3, 4\}$ as the two $x$-pairs that lie close together. For our choice of $C(x)$, the training objective to be maximized is:

$$\sum_{i,j \in A} \log p_G(x_i|E(x_j)) + \sum_{k,\ell \in B} \log p_G(x_k|E(x_\ell))$$
$$= \sum_{i,j \in A} \log p_G(x_i|z_j) + \sum_{k,\ell \in B} \log p_G(x_k|z_\ell) \tag{8}$$

The remainder of our proof is split into two cases:

**Case 1.** $\|z_j - z_\ell\| > \zeta$ for $j \in A, \ell \in B$

**Case 2.** $||z_j - z_\ell|| < \delta$ for $j \in A, \ell \in B$

Under Case 1, $x$ points that lie far apart also have $z$ encodings that remain far apart. Under Case 2, $x$ points that lie far apart have $z$ encodings that lie close together. We complete the proof by showing that the achievable objective value in Case 2 is strictly worse than in Case 1, and thus an optimal encoder/decoder pair would avoid the $x, z$ matching that leads to Case 2.

In Case 1 where $||z_j - z_\ell|| > \zeta$ for all $j \in A, \ell \in B$, we can lower bound the training objective (8) by choosing:

$$p_G(x_i|z_j) = \begin{cases} (1-\gamma)/2 & \text{if } i, j \in A \text{ or } i, j \in B \\ \gamma/2 & \text{otherwise} \end{cases} \tag{9}$$

with $\gamma = \sigma(-L\zeta) \in (0, \frac{1}{2})$, where $\sigma(\cdot)$ denotes the sigmoid function. Note that this ensures $\sum_{i \in [4]} p_G(x_i|z_j) = 1$ for each $j \in [4]$, and does not violate the Lipschitz condition from Assumption 2 since:

$$|\log p_G(x_i|z_j) - \log p_G(x_i|z_\ell)| \begin{cases} = 0 & \text{if } j, \ell \in A \text{ or } j, \ell \in B \\ \leq \log\left((1-\gamma)/\gamma\right) & \text{otherwise} \end{cases}$$

and thus remains $\leq L||z_j - z_\ell||$ when $\gamma = \sigma(-L\zeta) \geq \sigma(-L||z_j - z_\ell||) = 1/[1 + \exp(L||z_j - z_\ell||)]$.

Plugging the $p_G(x|z)$ assignment from (9) into (8), we see that an optimal decoder can obtain training objective value $\geq 8 \log [\sigma(L\zeta)/2]$ in Case 1 where $||z_j - z_\ell|| > \zeta, \forall j \in A, \ell \in B$.

Next, we consider the alternative case where $||z_j - z_\ell|| < \delta$ for $j \in A, \ell \in B$.

For $i, j \in A$ and for all $\ell \in B$, we have:

$$\begin{aligned} \log p_G(x_i|z_j) &\leq \log p_G(x_i|z_\ell) + L||z_j - z_\ell|| & \text{by Assumption 2} \\ &\leq \log p_G(x_i|z_\ell) + L\delta \\ &\leq L\delta + \log\left[1 - \sum_{k \in B} p_G(x_k|z_\ell)\right] & \text{since } \sum_k p_G(x_k|z_\ell) \leq 1 \end{aligned}$$

Continuing from (8), the overall training objective in this case is thus:

$$\sum_{i,j \in A} \log p_G(x_i|z_j) + \sum_{k,\ell \in B} \log p_G(x_k|z_\ell)$$

$$\leq 4L\delta + \sum_{i,j \in A} \min_{\ell \in B} \log\left[1 - \sum_{k \in B} p_G(x_k|z_\ell)\right] + \sum_{k,\ell \in B} \log p_G(x_k|z_\ell)$$

$$\leq 4L\delta + \sum_{\ell \in B}\left[2 \log\left(1 - \sum_{k \in B} p_G(x_k|z_\ell)\right) + \sum_{k \in B} \log p_G(x_k|z_\ell)\right]$$

$$\leq 4L\delta - 12 \log 2$$

using the fact that the optimal decoder for the bound in this case is: $p_G(x_k|z_\ell) = 1/4$ for all $k, \ell \in B$.

Finally, plugging our range for $\delta$ stated in the Theorem 2, it shows that the best achievable objective value in Case 2 is strictly worse than the objective value achievable in Case 1. Thus, the optimal encoder/decoder pair under the AAE with perturbed $x$ will always prefer the matching between $\{x_1, \ldots, x_4\}$ and $\{z_1, \ldots, z_4\}$ that ensures nearby $x_i$ are encoded to nearby $z_i$ (corresponding to Case 1). $\qquad\square$

## D    PROOF OF THEOREM 3

**Theorem 3.** *Suppose $x_1, \cdots, x_n$ are divided into $n/K$ clusters of equal size $K$, with $S_i$ denoting the cluster index of $x_i$. Let the perturbation process $C$ be uniform within clusters, i.e. $p_C(x_i|x_j) = 1/K$ if $S_i = S_j$ and $= 0$ otherwise. For a one-to-one encoder mapping $E$ from $\{x_1, \cdots, x_n\}$ to $\{z_1, \cdots, z_n\}$, the denoising objective $\max_{G \in \mathcal{G}_L} \frac{1}{n} \sum_{i=1}^n \sum_{j=1}^n p_C(x_j|x_i) \log p_G(x_i|E(x_j))$ is upper bounded by: $\frac{1}{n^2} \sum_{i,j:S_i \neq S_j} \log \sigma(L\|E(x_i) - E(x_j)\|) - \log K$.*

*Proof.* Without loss of generality, let $E(x_i) = z_i$ for notational convenience. We consider what is the optimal decoder probability assignment $p_G(x_i|z_j)$ under the Lipschitz constraint 2.

The objective of the AAE with perturbed $x$ is to maximize:

$$\frac{1}{n}\sum_i \sum_j p_C(x_j|x_i)\log p_G(x_i|E(x_j)) = \frac{1}{nK}\sum_j \sum_{i:S_i=S_j} \log p_G(x_i|z_j)$$

We first show that the optimal $p_G(\cdot|\cdot)$ will satisfy that the same probability is assigned within a cluster, i.e. $p(x_i|z_j) = p(x_k|z_j)$ for all $i, k$ s.t. $S_i = S_k$. If not, let $P_{sj} = \sum_{i:S_i=s} p_G(x_i|z_j)$, and we reassign $p_{G'}(x_i|z_j) = P_{S_ij}/K$. Then $G'$ still conforms to the Lipschitz constraint if $G$ meets it, and $G'$ will have a larger target value than $G$.

Now let us define $P_j = \sum_{i:S_i=S_j} p_G(x_i|z_j) = K \cdot p_G(x_j|z_j)$ $(0 \le P_j \le 1)$. The objective becomes:

$$\max_{p_G} \frac{1}{nK}\sum_j \sum_{i:S_i=S_j} \log p_G(x_i|z_j) = \max_{p_G} \frac{1}{n}\sum_j \log p_G(x_j|z_j)$$

$$= \max_{p_G} \frac{1}{n}\sum_j \log P_j - \log K$$

$$= \max_{p_G} \frac{1}{2n^2}\sum_i \sum_j (\log P_i + \log P_j) - \log K$$

$$\le \frac{1}{2n^2}\sum_i \sum_j \max_{p_G}(\log P_i + \log P_j) - \log K$$

Consider each term $\max_{p_G}(\log P_i + \log P_j)$: when $S_i = S_j$, this term can achieve the maximum value 0 by assigning $P_i = P_j = 1$; when $S_i \ne S_j$, the Lipschitz constraint ensures that:

$$\log(1 - P_i) \ge \log P_j - L\|z_i - z_j\|$$
$$\log(1 - P_j) \ge \log P_i - L\|z_i - z_j\|$$

Therefore:

$$\log P_i + \log P_j \le 2\log \sigma(L\|z_i - z_j\|)$$

Overall, we thus have:

$$\max_{p_G} \frac{1}{nK}\sum_j \sum_{i:S_i=S_j} \log p_G(x_i|z_j) \le \frac{1}{n^2}\sum_{i,j:S_i \ne S_j} \log \sigma(L\|z_i - z_j\|) - \log K$$

$\square$

# E  DATASETS

The Yelp dataset is from Shen et al. (2017), which has 444K/63K/127K sentences of less than 16 words in length as train/dev/test sets, with a vocabulary of 10K. It was originally divided into positive and negative sentences for style transfer between them. Here we discard the sentiment label and let the model learn from all sentences indiscriminately. Our second dataset of Yahoo answers is from Yang et al. (2017). It was originally document-level. We perform sentence segmentation and keep sentences with length from 2 to 50 words. The resulting dataset has 495K/49K/50K sentences for train/dev/test sets, with vocabulary size 20K.

# F  EXPERIMENTAL DETAILS

We use the same architecture to implement all models with different objectives. The encoder $E$, generator $G$, and the language model used to compute Forward/Reverse PPL are one-layer LSTMs with hidden dimension 1024 and word embedding dimension 512. The last hidden state of the encoder is projected into 128/256 dimensions to produce the latent code $z$ for Yelp/Yahoo datasets

respectively, which is then projected and added with input word embeddings fed to the generator. The discriminator $D$ is an MLP with one hidden layer of size 512. $\lambda$ of AAE based models is set to 10 to ensure the latent codes are indistinguishable from the prior. All models are trained via the Adam optimizer (Kingma & Ba, 2014) with learning rate 0.0005, $\beta_1 = 0.5$, $\beta_2 = 0.999$. At test time, encoder-side perturbations are disabled, and we use greedy decoding to generate $x$ from $z$.

## G    HUMAN EVALUATION

For the tense transfer experiment, the human annotator is presented with a source sentence and two outputs (one from each approach, presented in random order) and asked to judge which one successfully changes the tense while being faithful to the source, or whether both are good/bad, or if the input is not suitable to have its tense inverted. We collect labels from two human annotators and if they disagree, we further solicit a label from the third annotator.

## H    GENERATION-RECONSTRUCTION RESULTS ON THE YAHOO DATASET

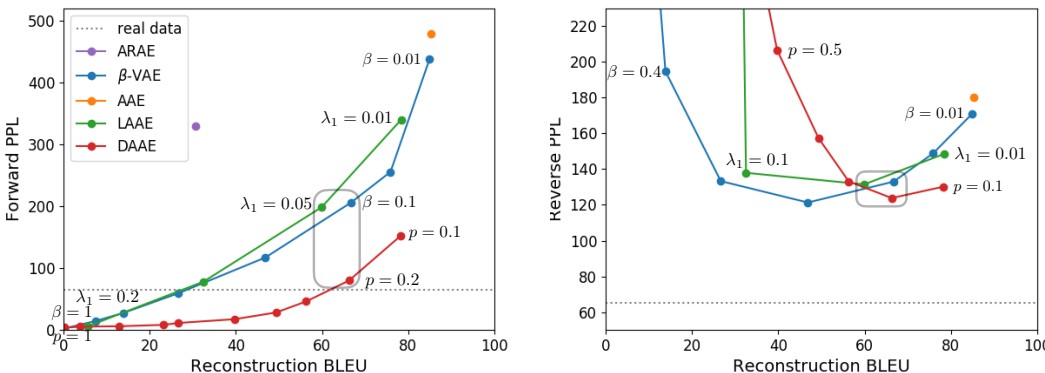

Figure H.1: Generation-reconstruction trade-off of various text autoencoders on Yahoo. The "real data" line marks the PPL of a language model trained and evaluated on real data. We strive to approach the lower right corner with both high BLEU and low PPL. The grey box identifies hyperparameters we use for respective models in subsequent experiments. Points of severe collapse (Reverse PPL > 300) are removed from the right panel.

## I    NEIGHBORHOOD PRESERVATION

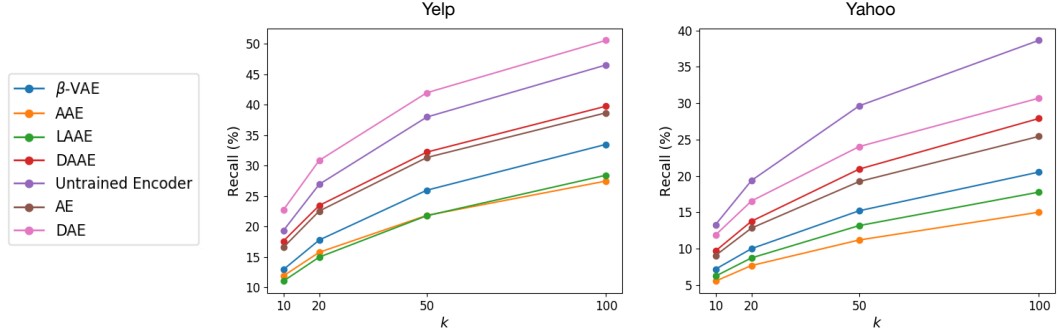

Figure I.2: Recall rate of 10 nearest neighbors in the sentence space retrieved by $k$ nearest neighbors in the latent space on the Yelp and Yahoo datasets. Here we include non-generative models AE and DAE. We find that an untrained RNN encoder from random initialization has a good recall rate, and we suspect that SGD training of vanilla AE towards only the reconstruction loss will not overturn this initial bias. Note that denoising still improves neighborhood preservation in this case. Also note that DAAE has the highest recall rate among all generative models that have a latent prior imposed.

## J    ADDITIONAL EXAMPLES

| Source | how many gospels are there that were n't included in the bible ? |
|---|---|
| 5-NN by AAE | there are no other gospels that were n't included in the bible . |
| | how many permutations are there for the letters in the word _UNK ' ? |
| | anyone else picked up any of the _UNK in the film ? |
| | what 's the significance of the number 40 in the bible ? |
| | how many pieces of ribbon were used in the _UNK act ? |
| 5-NN by DAAE | there are no other gospels that were n't included in the bible . |
| | how many litres of water is there in the sea ? |
| | how many _UNK gods are there in the classroom ? |
| | how many pieces of ribbon were used in the _UNK act ? |
| | how many times have you been grounded in the last year ? |
| Source | how do i change colors in new yahoo mail beta ? |
| 5-NN by AAE | how should you present yourself at a _UNK speaking exam ? |
| | how can i learn to be a hip hop producer ? |
| | how can i create a _UNK web on the internet ? |
| | how can i change my _UNK for female not male ? |
| | what should you look for in buying your first cello ? |
| 5-NN by DAAE | how do i change that back to english ? |
| | is it possible to _UNK a yahoo account ? |
| | how do i change my yahoo toolbar options ? |
| | what should you look for in buying your first cello ? |
| | who do you think should go number one in the baseball fantasy draft , pujols or _UNK ? |

Table J.1: Examples of nearest neighbors in the latent Euclidean space of AAE and DAAE on Yahoo dataset.

| Input | the staff is rude and the dr. does not spend time with you . | slow service , the food tasted like last night 's leftovers . |
|---|---|---|
| ARAE | the staff is rude and the dr. does not worth two with you . | slow service , the food tasted like last night 's leftovers . |
| $\beta$-VAE | the staff was rude and the dr. did not spend time with your attitude . | slow service , the food tastes like last place serves . |
| AAE | the staff was rude and the dr. does not spend time with you . | slow service , the food tasted like last night 's leftovers . |
| LAAE | the staff was rude and the dr. is even for another of her entertained . | slow service , the food , on this burger spot ! |
| DAAE | the staff was rude and the dr. did not make time with you . | slow service , the food tastes like last night ... . |
| | | |
| Input | they are the worst credit union in arizona . | i reported this twice and nothing was done . |
| ARAE | they are the worst bank credit in arizona . | i swear this twice and nothing was done . |
| $\beta$-VAE | they were the worst credit union in my book . | i 've gone here and nothing too . |
| AAE | they are the worst credit union in arizona . | i reported this twice and nothing was done . |
| LAAE | they were the worst credit union in my heart . | i dislike this twice so pleasant guy . |
| DAAE | they were the worst credit union in arizona ever . | i hate this pizza and nothing done . |

Table J.2: Additional examples of vector arithmetic for tense inversion.

|  | AAE | DAAE |
|---|---|---|
| **Input** | **this woman was extremely rude to me .** | **this woman was extremely rude to me .** |
| $+v$ | this woman was extremely rude to me . | this woman was extremely nice . |
| $+1.5v$ | this woman was extremely rude to baby . | this staff was amazing . |
| $+2v$ | this woman was extremely rude to muffins . | this staff is amazing . |
| **Input** | **my boyfriend said his pizza was basic and bland also .** | **my boyfriend said his pizza was basic and bland also .** |
| $+v$ | my boyfriend said his pizza was basic and tasty also . | my boyfriend said his pizza is also excellent . |
| $+1.5v$ | my shared said friday pizza was basic and tasty also . | my boyfriend and pizza is excellent also . |
| $+2v$ | my shared got pizza pasta was basic and tasty also . | my smoked pizza is excellent and also exceptional . |
| **Input** | **the stew is quite inexpensive and very tasty .** | **the stew is quite inexpensive and very tasty .** |
| $-v$ | the stew is quite inexpensive and very tasty . | the stew is quite an inexpensive and very large . |
| $-1.5v$ | the stew is quite inexpensive and very very tasteless . | the stew is quite a bit overpriced and very fairly brown . |
| $-2v$ | the – was being slow - very very tasteless . | the hostess was quite impossible in an expensive and very few customers . |
| **Input** | **the patrons all looked happy and relaxed .** | **the patrons all looked happy and relaxed .** |
| $-v$ | the patrons all looked happy and relaxed . | the patrons all helped us were happy and relaxed . |
| $-1.5v$ | the patrons all just happy and smelled . | the patrons that all seemed around and left very stressed . |
| $-2v$ | the patrons all just happy and smelled . | the patrons actually kept us all looked long and was annoyed . |

Table J.3: Additional examples of vector arithmetic for sentiment transfer.

| Input 1 | **what language should i learn to be more competitive in today 's global culture ?** |
|---|---|
| Input 2 | **what languages do you speak ?** |
| AAE | what language should i learn to be more competitive in today 's global culture ? |
|  | what language should i learn to be more competitive in today 's global culture ? |
|  | what language should you speak ? |
|  | what languages do you speak ? |
|  | what languages do you speak ? |
| DAAE | what language should i learn to be more competitive in today 's global culture ? |
|  | what language should i learn to be competitive today in arabic 's culture ? |
|  | what languages do you learn to be english culture ? |
|  | what languages do you learn ? |
|  | what languages do you speak ? |
| **Input 1** | **i believe angels exist .** |
| **Input 2** | **if you were a character from a movie , who would it be and why ?** |
| AAE | i believe angels exist . |
|  | i believe angels - there was the exist exist . |
|  | i believe in tsunami romeo or <unk> i think would it exist as the world population . |
|  | if you were a character from me in this , would we it be ( why ! |
|  | if you were a character from a movie , who would it be and why ? |
| DAAE | i believe angels exist . |
|  | i believe angels exist in the evolution . |
|  | what did <unk> worship by in <unk> universe ? |
|  | if you were your character from a bible , it will be why ? |
|  | if you were a character from a movie , who would it be and why ? |

Table J.4: Interpolations between two input sentences generated by AAE and our model on the Yahoo dataset.

