# OpenReview forum: "Denoising Improves Latent Space Geometry in Text Autoencoders"
_ICLR.cc/2020/Conference — Reject_

### Official Review · AnonReviewer2 · 2019-10-19
**Official Blind Review #2**

**Rating:** 3

**Review:**

This paper presented a denoising adversarial autoencoder for sentence embeddings. The idea is that by introducing perturbations (word omissions, etc) the embeddings are more meaningful and less "memorized". Evaluations include measuring sentence perplexity in generation/reconstruction, tense changing via vector arithmetic, sentiment changes via negative/positive vector additions, and sentence interpolations.

Strengths: I thought the idea is nice, and the results do seem to show improvements in a number of interesting tasks.

Weaknesses: I don't really think the explanation, especially in Theorem 1, makes a lot of sense. Qualitatively speaking, it's true that "memorization" in autoencoders (where the latent space has a 1-1 mapping with the input space) is problematic when the autoencoders are too powerful, but it is not always the case, and it is too far to say that the probability in theorem 1 is ALWAYS agnostic to encoding. The fact is word2vec works just fine with no perturbations, and there is no mathematical reason why sentence embeddings are fundamentally different. What is more accurate to say is that there is a tradeoff between model complexity and latent space representation usefulness, which is also related to the regularization/overfitting tradeoff in supervised learning. Here, injecting noise in the exact same fashion proposed in this paper is a well-accepted practice. While I think it's interesting that it works well here, I wouldn't really frame it as such a novelty, in that case, and I believe the other works on denoising autoencoders should be compared against in the experiments. In general, I find the mathematical claims a bit dubious, as a main assumption seems to be that the autoencoder itself is so overparametrized that it isn't really functioning as a representation-learning tool anyway.

I also feel that the last experiment (referenced in the appendix) needs to go in the main text if we're to see it as a contribution. There is some wording that can be tightened in the main text, to make more room.

Overall, I would improve my rating if the paper refocused more on the experiments, included more baselines (like other denoising autoencoders) and tasks that measure something besides perplexity (such as actual sentiment prediction, or machine translation, or other somewhat unrelated downstream tasks), and decreased the emphasis on the theoretical analysis--unless there is something I am significantly misunderstanding, it does not seem to be a particularly powerful theoretical contribution.

**Experience Assessment:**

I have read many papers in this area.

**Review Assessment: Checking Correctness Of Derivations And Theory:**

I assessed the sensibility of the derivations and theory.

**Review Assessment: Checking Correctness Of Experiments:**

I assessed the sensibility of the experiments.

**Review Assessment: Thoroughness In Paper Reading:**

I read the paper at least twice and used my best judgement in assessing the paper.

---

> ### Author Response · Authors · 2019-11-10
> **Response to Reviewer#2**
>
> We thank the reviewer for the feedback. We would like to clarify that the focus of this paper is *controllable text generation*, and we study autoencoder based text generative models as a tool that can manipulate text via latent vector operations. Therefore, we conduct experiments on various text generation tasks, and we compare our method to latent variable generative models but not to DAEs which cannot be employed generatively. We will update our paper to make this point clear.
>
> The reviewer raised concerns about the use of over-parameterized autoencoders for representation learning. We would like to note that for controllable text generation, we actually need flexible models that have high content fidelity after encoding-decoding, while enforcing the latent codes to be Gaussian. In our experiments, the dimension of the latent variable z is not large (128 for Yelp dataset and 256 for Yahoo dataset), but the encoder and decoder networks are large in order to achieve good performance (we used 1024 LSTM hidden dimension and 512 word embedding dimension, the same network size is used in Kim et al. (2018)). For all models, we observed that smaller networks produced worse empirical results. In fact, the top performing text representation models are heavily over-parameterized (Devlin et al., 2018). Extensive empirical observations on a wide range of tasks have shown that over-parameterization of deep neural networks does not lead to overfitting but rather improves generalization, a phenomenon that has attracted a lot of research interest (Zhang et al., 2017; Neyshabur et al., 2019). Furthermore, recent research suggests that what matters is not the raw capacity of the model, but rather the "simplicity" of the learned function, and that higher-capacity models may (counterintuitively) learn simpler functions in practice (Belkin et al., 2018).
>
> This is why we theoretically analyze powerful encoder/decoder networks, as these are the text models used in practice. Note that our theory is *not* stating that the 1-1 mapping / memorization of AAE is problematic (we will update our paper to clarify this). In fact, we require an (approximately) 1-1 mapping in order to handle fine-grained controllable generation. Since there exist many such x-z mappings, the question we address is which type of mapping (with what geometric properties) will the autoencoder learn. Here, we are able to prove that in terms of global optimality, DAAE will specifically learn only the x-z mapping in which x-neighborhoods are preserved in the z-space, whereas AAE has no such guarantees and may learn an arbitrary x-z mapping while still achieving optimality of its training objective. The theoretical insights provided by our analysis are also reflected in practice: Sec 5.2 verified that, consistent with our theory, DAAE has the best neighborhood preservation property in the real-world case.
>
> We’d like to emphasize that our main contribution is not only the proposed model, but also the theoretical & empirical analysis that this simple use of denoising is truly superior. While denoising is a well-accepted practice, there is little understanding of its impact. Previous analyses of denoising only drew intuitive connections to manifold learning and more robust representations (Vincent et al., 2008; Bengio et al., 2013). We first present rigorous mathematical explanations of how denoising induces geometric latent space organization in text autoencoders and why it is superior to no denoising. Our theory is verified empirically. Thanks to the improved latent space geometry by denoising, we have successfully achieved not only good perplexity results, but also respectable performance on controllable text generation tasks for the first time from completely unsupervised data, including style transfer via vector arithmetic and sentence interpolation via latent space traversal. These completely unsupervised applications in our paper are novel as well.
>
> Finally, we’d like to note that for downstream classification tasks, BERT-style models have an overwhelming advantage over approaches that encode sentences into a single vector. In contrast, for generative modeling, it is easy to impose a latent prior on single-vector representations and manipulate them, but it is much more difficult to impose a prior on and manipulate variable-length vector sequences. That’s why we didn’t use text autoencoders to compete in classification tasks. Our experiments focus on generation, and many important applications from summarization to style transfer are generation tasks.
>
> - add the last experiment into main text
> We will include this last experiment into the main text in the revised version of the paper that will be uploaded shortly.
>
> We hope through the above discussions the reviewer can reassess this work. We will improve clarity in the paper, and we are always available for further discussions.

---

> > ### Author Response · Authors · 2019-11-10
> > **(continued)**
> >
> > References:
> >
> > Kim et al. (2018) Semi-Amortized Variational Autoencoders
> >
> > Devlin et al. (2018) BERT: Pre-training of Deep Bidirectional Transformers for Language Understanding
> >
> > Zhang et al. (2017) Understanding deep learning requires rethinking generalization
> >
> > Neyshabur et al. (2019) Towards understanding the role of over-parametrization in generalization of neural networks
> >
> > Belkin et al. (2018) Reconciling modern machine learning practice and the bias-variance trade-off
> >
> > Vincent et al. (2008) Extracting and Composing Robust Features with Denoising Autoencoders
> >
> > Bengio et al. (2013) Generalized Denoising Auto-Encoders as Generative Models

---

> > > ### Comment · AnonReviewer2 · 2019-11-14
> > > **idea is fine, writing it as a theorem seems a stretch.**
> > >
> > > I think I better understand what you are trying to say in theorem 1 and 2 but I still do not believe they are mathematically rigorous.
> > >
> > > Say my set X = {1,2} and Z = {-1,1}. I can construct an encoder E1(1) = E1(2) = -1 and another encoder E2(1)=1, E2(2) = -1. Then your statement is false and both are valid lipschitz encoders (for a large enough lipschitz constant.)
> > >
> > > Theorem 2: the assumption statement doesn't make sense. What if I have 4 points xk all within epsilon of xi; then they can't all have probability 1/2 (assuming discrete X).
> > >
> > > In fact I think there may be much more fundamental results in information theory that basically say what you are trying to convey, that when the encoder is constrained to maintain some kind of coherence between neighboring inputs, its choices of outputs are more limited.

---

> > > > ### Author Response · Authors · 2019-11-15
> > > > **Thank you for your feedback**
> > > >
> > > > We thank the reviewer for the feedback. While the results hold, we realize that the current writing of the theorems may be unclear/misleading. We clarify below and will revise our paper accordingly:
> > > >
> > > > - Theorem 1
> > > > We meant “For any *one-to-one* encoder mapping E from {x1 ... xn} to {z1 … zn}, the optimal value of objective … is the same”. Here we analyze which x-z mapping will the encoder/decoder learn under global optimality. If the encoder maps different x to the same z, the reconstruction loss will be strictly worse and an optimal AAE will not learn such mappings. The remaining question is which one-to-one mapping AAE will learn, which is what Theorem 1 studies.
> > > >
> > > > In the reviewer’s example, the second encoder E2 would always be favored over E1 since E2 enables better reconstruction. (Also note that we have not assumed the encoder/decoder are Lipschitz on discrete input x, we only assume that the decoder is Lipschitz on its continuous input z.)
> > > >
> > > > - Theorem 2
> > > > The context in which the theorem applies is described in the previous paragraph: “there are two pairs of x closer together and also two pairs of z closer together”. Here, the perturbation probability can be p_C(xi | xj) = 1/2 if d(xi, xj) < eps and = 0 otherwise. For improved clarity, we’ll move this setup context inside of the statement of Theorem 2.
> > > >
> > > > - “there may be much more fundamental results in information theory that basically say what you are trying to convey, that when the encoder is constrained to maintain some kind of coherence between neighboring inputs, its choices of outputs are more limited.”
> > > >
> > > > We agree it would be nice to have a broader information theoretic analysis of the encodings of discrete sequences. However, we are not aware of any specific references that would address the issue we are studying, especially how the use of input-denoising can help impose a particular geometry on the resulting encodings.

---

### Official Review · AnonReviewer1 · 2019-10-21
**Official Blind Review #1**

**Rating:** 6

**Review:**

The paper "Denoising Improves Latent Space Geometry in Text Autoencoders" tackles the problem of text autoencoding in a space which respects text similarities. It is an interesting problem for which various attempts have been proposed, while still facing difficulties for encoding in smooth spaces. The paper proposes a simple (rather straightforward) approach based on adversarial learning, with some theoretical guarantees, which obtains good performances for reconstruction and neighborhood preservation.

My main concern is about the missing of comparison with word dropout with variational encoding [Bowman et al., 2016], which also considers perturbations of the input texts to enforce the decoder to use the latent space. While the authors cite this work, I cannot understand why they did not include it in their experiments.

Also, theorem 3 gives an upperbound of the achievable log-likelihood, which is "substantially better when examples in the same cluster are mapped to to points in the latent space in a manner that is well-separated from encodings of other
clusters". Ok but what does it show for the approach. If it was a lower-bound of the DAAE likelihood it would be interesting. But an upperbound ? In which sense does it indicate that it will be better than AAE ? Wouldn't it be possible to theoretically analyze other baselines? Also, all the theoretical analysis is made based on strong assumptions. Are these verified on considered datasets?


Minor questions :
          - In introduction of the experiments section, authors mention that they tried word removal and word masking. What is the difference ?
           - what is the language model used for forward and reverse ppl ?



**Experience Assessment:**

I have published one or two papers in this area.

**Review Assessment: Checking Correctness Of Derivations And Theory:**

I assessed the sensibility of the derivations and theory.

**Review Assessment: Checking Correctness Of Experiments:**

I assessed the sensibility of the experiments.

**Review Assessment: Thoroughness In Paper Reading:**

I read the paper at least twice and used my best judgement in assessing the paper.

---

> ### Author Response · Authors · 2019-11-07
> **Response to Reviewer#1**
>
> We thank the reviewer for the feedback. We address each question in turn and will add the clarifications in the revision:
>
> - VAE with word dropout on the decoder side (Bowman et al., 2016)
>
> Bowman et al. proposed to weaken VAE’s decoder by masking words on the decoder side to help alleviate its collapse issue. However, as the authors pointed out in their paper: “Even with the techniques described in the previous section, including the inputless decoder, we were unable to train models for which the kl divergence term of the cost function dominates the reconstruction term”. From Table 2 in their paper, we can see that the VAE with inputless decoder has a small KL term (15) and a large reconstruction loss (120-15=105), which indicates that the latent z encodes little information about x and it cannot do reconstruction well. We also tried it in our experiments. On the Yelp dataset, the best reconstruction BLEU it can achieve is 12.8 with word dropout rate=0.7 (our model has BLEU 84.3). Therefore, it is not suitable for text manipulations that require high content fidelity.
>
> - Theorem 3 gives an upper bound, but what does it show?
>
> We will clarify that the goal of our analysis is not to compare the objective values of AAE vs DAAE.  Instead, we want to analyze which x-z mapping the model will learn under the AAE and DAAE objective. With powerful encoder/decoder networks, AAE has no preference over different x-z mappings because they can all achieve the same optimal objective value (Thm 1). In contrast, DAAE prefers organized x-z mappings (that preserve local neighborhoods in the x-space) over disorganized ones, since organized mappings can achieve better objective values (Thm 2, 3). In conclusion, a well-trained DAAE is guaranteed to learn neighborhood-preserving latent representations, whereas even a perfectly-trained AAE model may learn latent representations whose geometry fails to reflect similarity in the x space.
>
> We agree with the reviewer that we’d ideally like to derive both upper and lower bounds on the achievable DAAE objective value. Nevertheless, the upper bound provided by Thm 3 implies that an organized x-z mapping has a better achievable limit than a disorganized mapping, thus supporting our argument that the denoising criterion will encourage a better geometrically-organized latent space (Sec 5.2 empirically verified this theoretical conclusion).
>
> - Assumptions made by our theoretical analysis
>
> The assumptions we made are:
> (1) an effectively trained discriminator ensures that the latent codes resemble samples from the prior;
> (2) the encoder and decoder are high-capacity models that are universal approximators, with the constraint that
> (3) the decoder is Lipschitz continuous in its continuous input z.
>
> For (1):  In all the experiments we did, training was very stable and the adversarial loss was kept at around -log 0.5, indicating that the latent codes were indistinguishable from the prior and this assumption holds empirically.
> For (2):  Schäfer and Zimmermann (2006) have shown the universal approximation ability of RNNs, and nowadays most state of the art sequence models employ high-capacity autoregressive neural networks with tons of parameters (Radford et al., 2019). In fact, the empirical ability of neural decoders approximate arbitrary distributions is widely cited as a reason for posterior collapse in recurrent VAEs (Chen et al., 2017; van den Oord et al., 2018; Dieng et al., 2018; Razavi et al., 2019), and thus this assumption is supported by a wealth of recent literature.
> For (3): This is a very weak assumption (also made in Mueller et al., 2017), and it holds as long as the recurrent or attention weight matrices in RNN/Transformer have bounded norm, which is naturally encouraged by SGD training with early stopping and L2 regularization (Zhang et al., 2017).
>
> - To theoretically analyze other baselines
>
> Our analysis applies to AAE-based models where the latent prior is imposed by an adversarial discriminator. To analyze the latent space geometry of (beta-)VAE would require different techniques, which is not the focus of this paper.
>
> There are many interesting open questions in this area, and we would like to emphasize that this is the first theoretical analysis of how denoising induces latent space organization in text autoencoders. Existing analyses of denoising autoencoders (Vincent et al., 2008; Bengio et al., 2013) informally argued that denoising can help learn data manifolds and extract more robust representations, but did not notice that a major benefit of denoising is that it encourages the preservation of data structure in the latent space (regardless if the data live on a manifold or not).

---

> > ### Author Response · Authors · 2019-11-07
> > **(continued)**
> >
> > - Difference between word removal and word masking
> >
> > Word removal will remove words from the sentence. The resulting sentence has no placeholder for the removed words and is shorter in length. In contrast, word masking will replace words with a <mask> token and preserve sentence length. For example, for the sentence “we had a very nice experience”, after removing “very” it becomes “we had a nice experience”, and after masking “very” it becomes “we had a <mask> nice experience”.
> >
> > - What is the language model used for forward and reverse ppl ?
> >
> > We used a LSTM language model which has one layer, 1024 hidden dimension and 512 word embedding dimension.
> >
> >
> > References:
> >
> > Schäfer and Zimmermann (2006). Recurrent Neural Networks Are Universal Approximators. https://link.springer.com/chapter/10.1007/11840817_66
> >
> > Radford et al. (2019). Language Models are Unsupervised Multitask Learners. https://openai.com/blog/better-language-models/
> >
> > Chen et al. (2017). Variational Lossy Autoencoder. https://arxiv.org/abs/1611.02731
> >
> > van den Oord et al. (2018). Neural Discrete Representation Learning. https://arxiv.org/abs/1711.00937
> >
> > Dieng et al. (2018). Avoiding Latent Variable Collapse with Generative Skip Models. https://arxiv.org/abs/1807.04863
> >
> > Razavi et al. (2019).  Preventing Posterior Collapse with delta-VAEs. https://openreview.net/pdf?id=BJe0Gn0cY7
> >
> > Mueller et al. (2017). Sequence to better sequence: continuous revision of combinatorial structures. http://proceedings.mlr.press/v70/mueller17a.html
> >
> > Zhang et al. (2017). Understanding deep learning requires rethinking generalization. https://arxiv.org/abs/1611.03530
> >
> > Vincent et al. (2008). Extracting and Composing Robust Features with Denoising Autoencoders. https://www.cs.toronto.edu/~larocheh/publications/icml-2008-denoising-autoencoders.pdf
> >
> > Bengio et al. (2013). Generalized Denoising Auto-Encoders as Generative Models. https://arxiv.org/pdf/1305.6663.pdf

---

### Official Review · AnonReviewer3 · 2019-10-22
**Official Blind Review #3**

**Rating:** 6

**Review:**

The paper argues that adding noise to the inputs of an adversarial autoencoder for text improves the geometry of the learned latent space (in terms of mapping similar input sentences to nearby points in the latent space). The authors present a mathematical argument for why adding noise to the inputs would enforce latent space structure while a vanilla autoencoder would have no preference over x-z mappings.

Overall, the paper addresses an important problem of improving autoencoder based generative models of text, presents a simple solution to do so and mathematically and empirically demonstrates its effectiveness. While I think that the benchmarks are somewhat artificial with small sentences and vocabulary sizes, I think the improvements demonstrated are substantial enough.

I have a few questions & comments

1) I’m curious about whether the authors have an intuition for why input space noise is better than latent space noise? Poole et al 2014 [1] showed that additive latent space gaussian noise in autoencoders is equivalent to a contractive autoencoder penalty and contractive autoencoders have an *explicit* penalty to encourage minimal change in z when changing x (i.e.) penalizing the norm of ||dz/dx||. Additive latent space noise appears to be a key ingredient to getting the ARAE and similar work like in Subramanian et al 2018 [2] to work. Was the LAAE implemented in the same framework as your DAAE?
2) It would be great to see Forward / Reverse PPL results on bigger datasets like the BookCorpus or WMT similar to [2].
3) You may be able to get similar reconstruction vs sample quality trade-offs with ARAEs by varying the variance of the gaussian noise, similar to LAAEs.
4) In Figure 3, I would really like to see how an autoencoder that isn’t a generative model performs. How well would a vanilla autoencoder or vanilla DAE perform? This is a cool setup to evaluate latent space representation quality - you could even consider running some of the SentEval probing tasks on these representations.
5) Could you use something like gradient-based latent space walks like in [2] to characterize the latent space geometry? https://arxiv.org/abs/1711.08014 also use similar gradient-based walks to characterize latent space smoothness in deep generative models. For example, if it takes 10 latent space gradient steps with a fixed learning rate for model “a” to turn sentence “x” into a *similar* sentence “y” but 20 steps for model “b”, then maybe “a” has smoother latent space geometry.
6) Theorem 1 - What if the set of zs isn’t unique and there is some sort of encoder collapse? Does this theorem still hold? (i.e.) there exists some set of points x_1, x_2 .. x_i \in x, that all map to z_k (and even potentially in the limit that all points in x map to the same point in z space).
7) It would be good to point out that the model presented in this work is far from SOTA on sentiment style transfer benchmarks like Yelp.

[1] Analyzing noise in autoencoders and deep networks - https://arxiv.org/pdf/1406.1831.pdf
[2] Towards Text Generation with Adversarially Learned Neural Outlines - https://papers.nips.cc/paper/7983-towards-text-generation-with-adversarially-learned-neural-outlines.pdf

**Experience Assessment:**

I have published one or two papers in this area.

**Review Assessment: Checking Correctness Of Derivations And Theory:**

I assessed the sensibility of the derivations and theory.

**Review Assessment: Checking Correctness Of Experiments:**

I assessed the sensibility of the experiments.

**Review Assessment: Thoroughness In Paper Reading:**

I read the paper at least twice and used my best judgement in assessing the paper.

---

> ### Author Response · Authors · 2019-11-13
> **Response to Reviewer#3**
>
> We thank the reviewer for the feedback and comments. We address each of them in turn and will make the corresponding clarifications in the paper.
>
> 1) “why input space noise is better than latent space noise? Poole et al 2014 [1] showed that additive latent space gaussian noise in autoencoders is equivalent to a contractive autoencoder penalty and contractive autoencoders have an *explicit* penalty to encourage minimal change in z when changing x (i.e.) penalizing the norm of ||dz/dx||.”
>
> Poole et al. (2014) studied continuous x, their autoencoder was a simple one-layer network with tied weight matrix, and their loss was squared reconstruction error. Namely, their encoder was h=f(Wx) with a single element-wise non-linearity, and the decoder was linear x_hat = W’h. In this setting, they showed that adding noise to h with a variance according to the encoder Jacobian can recover the contractive Jacobian norm regularization penalty. This relies heavily on the linearity of reconstruction, squared error and continuity of x, and does not really apply to our case where the encoder/decoder are complex models and x is discrete.
>
> When x is discrete and the encoder is complex, adding Gaussian noise to latent encodings no longer connects back to simple changes in the input text. Instead, denoising can directly control the latent encodings of perturbed versions of x by asking them to decode back to the original x. As our theorems show, it is advantageous for these latent vectors to geometrically concentrate so as to help a z-continuous decoder map these sets to common targets. This also encourages the decoder to treat the perturbations as related and share some of the generative process.
>
> “Was the LAAE implemented in the same framework as your DAAE?”
> Yes, we implemented all models in the same framework, just with different objectives.
>
> 2) “Forward / Reverse PPL results on bigger datasets like the BookCorpus or WMT”
>
> We conducted extensive experiments on the benchmark datasets of text autoencoders to thoroughly investigate their generation-reconstruction trade-off. We agree with the reviewer that better performance for text manipulation can be achieved by training larger models on larger datasets. We would like to investigate this in future work.
>
> 3) “You may be able to get similar reconstruction vs sample quality trade-offs with ARAEs by varying the variance of the gaussian noise, similar to LAAEs.”
>
> Thank you for the suggestion. We tried injecting latent noise into ARAE, but its generation-reconstruction trade-off curve is strictly worse than LAAE, so we only included the original ARAE as in their paper. Cífka et al. (2018) also reported similar findings that AAE is superior to ARAE.
>
> 4) “In Figure 3... How well would a vanilla autoencoder or vanilla DAE perform?”
>
> AE’s recall rate is slightly lower than DAAE, and DAE’s recall rate is the highest among all models. We found that an untrained RNN encoder from random initialization has a good recall rate (the ranking is DAE > untrained encoder > DAAE > AE > the rest), and we suspect that SGD training of vanilla AE towards only the reconstruction loss will not overturn this initial bias. Note that denoising still improves neighborhood preservation in this case.
>
> That said, we believe that considering primarily generative models is a fair comparison and is most consistent with the main text. When a latent prior is included/enforced for generative purpose, the mappings learned by the model have different properties. We’ll nevertheless include the figure that includes non-generative AE, DAE and untrained encoder in the appendix.
>
> 5) “gradient-based latent space walks”
>
> We agree with the reviewer that using gradient-based latent space walks is an interesting setup. It is, however, a bit different from our goals. Our paper studies whether it is possible to learn simple latent space geometry to map similar x to similar z and manipulate x via linear latent vector arithmetic. While taking gradient steps with respect to the decoder would enable complex non-linear interpolating trajectories (imposing implicit decoder geometry on the latent space), it wouldn’t quantify whether a simple encoded latent space geometry was achieved. Even models with complex latent geometries may still be able to move from one sentence to another within a few gradient steps. The linked work of Shao et al. specifically uses these gradient steps to gauge (manifold) geometry of the inputs themselves, whereas our analyses aims to gauge how well the latent space of different models reflects structure in the data space. Similarly, the cited work of Subramanian et al. uses these gradient steps as a way to manage poorly-structured latent spaces, whereas our goal is to analyze the geometry of the latent spaces, regardless of how poorly-structured they may be for certain models.

---

> > ### Author Response · Authors · 2019-11-13
> > **(continued)**
> >
> > 6) “Theorem 1 - What if the set of zs isn’t unique and there is some sort of encoder collapse? Does this theorem still hold? (i.e.) there exists some set of points x_1, x_2 .. x_i \in x, that all map to z_k (and even potentially in the limit that all points in x map to the same point in z space).”
> >
> > Theorem 1 analyzes which type of x-z mapping the AAE model will learn when it has achieved global optimality of its training objective. When global optimality is achieved, the discriminator will prevent all points in x from being mapped to the same point in z space because they must be indistinguishable from Gaussian. Moreover, different x will be encoded to different z for the decoder to best reconstruct them. (In practice, we have never observed encoder collapses in AAE that map different x to the same z, in contrast to VAE-variants).
> >
> > 7) “the model presented in this work is far from SOTA on sentiment style transfer benchmarks like Yelp.”
> >
> > Our model is trained in a fully unsupervised manner (no sentiment labels are provided during training), and at test time it can perform various style transfers by adding simple fixed offset vectors. We agree that our model is less powerful than SOTA sentiment transfer models that are specifically trained with labeled data.  We will add a note for this, also highlighting that the unsupervised task our model is used for is more challenging.  Note also that the term “unsupervised” has different senses in the literature pertaining to sentiment transfer. E.g., the method of Yang et al. view the task as “unsupervised” even though sentiment-information is used during training.
> >
> > References:
> >
> > Cífka et al. (2018). “Eval all, trust a few, do wrong to none: Comparing sentence generation models”. https://arxiv.org/pdf/1804.07972.pdf
> >
> > Shao et al. (2017). “The Riemannian Geometry of Deep Generative Models”. https://arxiv.org/pdf/1711.08014.pdf
> >
> > Subramanian et al. (2018). “Towards Text Generation with Adversarially Learned Neural Outlines”. https://papers.nips.cc/paper/7983-towards-text-generation-with-adversarially-learned-neural-outlines.pdf
> >
> > Yang et al. (2019). “Unsupervised Text Style Transfer using Language Models as Discriminators”. https://arxiv.org/pdf/1805.11749.pdf

---

> > > ### Author Response · Authors · 2019-11-15
> > > **Theorem 1**
> > >
> > > As Reviewer#2 also pointed out, in Theorem 1, we actually meant “For any *one-to-one* encoder mapping E from {x1 ... xn} to {z1 … zn}, the optimal value of objective … is the same” (we’re revising our paper to correct this). If the encoder maps different x to the same z, the reconstruction loss will be strictly worse and an optimal AAE will not learn such mappings. The remaining question is which one-to-one mapping AAE will learn, which is what Theorem 1 studies.

---

### Author Response · Authors · 2019-11-15
**Revision is uploaded**

We uploaded a revision that has addressed the reviewers' comments. We thank all reviewers for their useful feedback, and we believe that the clarity and completeness of our paper has improved through discussion.

---

### Decision · Program_Chairs · 2019-12-19

**Decision:**

Reject

**Comment:**

This work presents a simple technique for improving the latent space geometry of text autoencoders. The strengths of the paper lie in the simplicity of the method, and results show that the technique improves over the considered baselines. However, some reviewers expressed concerns over the presented theory for why input noise helps, and did not address concerns that the theory was useful. The paper should be improved if Section 4 were instead rewritten to focus on providing intuition, either with empirical analysis, results on a toy task, or clear but high level discussion of why the method helps. The current theorem statements seem either unnecessary or make strong assumptions that don't hold in practice. As a result, Section 4 in its current form is not in service to the reader's understanding why the simple method works.
Finally, further improvements to the paper could be made with comparisons to additional baselines from prior work as suggested by reviewers.